# Comparison of Simulated and Measured Power of the Earth-Space Link for Satellite-Based AIS Signals

**DOI:** 10.3390/s23156740

**Published:** 2023-07-27

**Authors:** Xiang Dong, Zhigang Yuan, Fang Sun, Qinglin Zhu, Mingchen Sun, Pengfei Zhu

**Affiliations:** 1School of Electronic Information, Wuhan University, Wuhan 430072, China; y_zgang@whu.edu.cn; 2China Research Institute of Radiowave Propagation, Qingdao 266107, China; sf427@163.com (F.S.); zhuql1@crirp.ac.cn (Q.Z.); sunmc@crirp.ac.cn (M.S.); zhupf@crirp.ac.cn (P.Z.)

**Keywords:** AIS, ionosphere, faraday rotation, Tiantuo-5, TEC, VHF, UHF

## Abstract

This research aims to analyze the impact of the Earth-Space link on the Automatic Identification System (AIS) signals of ships. To achieve this, we established a simulation system that measures the receiving power of AIS signals via satellite platforms. We validated the system by utilizing observation data from Tiantuo-5. Through this simulation, we quantitatively analyzed the effects of ionospheric TEC (Total Electron Content) and space loss on the received power. During the processing of observation data, we construct a geometric propagation model utilizing the measured positions of both the satellite and the ship. We then calculate the antenna gain and remove any system errors. Additionally, we eliminate the deviation of elevation and azimuth angles caused by satellite motion. This allows us to determine the actual power of different ships reaching the receiving platform. Upon comparing the measured power data with the simulated power, it was noted that both exhibited an increasing trend as the elevation angle increased. This led to an RMSE (Root Mean Square Error) result of approximately one, indicating the accuracy of the simulation system. These findings hold significant implications for analyzing interference factors in satellite-ground links.

## 1. Introduction

The Automatic Identification System (AIS) utilizes open broadcast technology to regularly transmit a ship’s position, speed, and other pertinent information. Operating within the VHF maritime band, AIS facilitates simple tracking, identification, and exchange of navigation data between vessels or with the shore. However, due to the Earth’s curvature, the communication range for ship-to-ship and ship-to-shore is limited to a maximum distance of 60 nautical miles. Satellite-based AIS technology offers a promising solution to overcome the limitations of terrestrial VHF (Very High Frequency) coverage and effectively cover any area on Earth [1,2,3]. In recent years, the development of AIS signal reception systems based on satellite platforms has progressed rapidly, enabling the real-time reception of AIS signals worldwide. The satellite receives the AIS signals on the VHF uplink, making it a reliable and efficient option for global coverage. The AIS information is stored and transmitted to the ground station through either the S-band channel or the UHF (Ultra High Frequency) downlink available on the satellite. The S-band channel allows for high-speed data transmission [4,5,6]. However, the AIS signals must pass through the Earth’s troposphere and ionosphere atmosphere from ships to satellites [7,8]. The atmosphere can alter the signal characteristics at different locations and times, particularly when influenced by the ionosphere. Receiving AIS signals in space presents several challenges. The first challenge is message collision due to the large antenna footprint of the AIS satellite. The second challenge is low signal strength. Additionally, the CubeSat design concept imposes strict limitations on cost, available power, and antenna design, with size being a particular constraint [9,10]. To overcome these obstacles and receive AIS messages from space, a satellite-platform AIS signal reception power simulation system is necessary.

In this work, a satellite platform AIS signal reception power simulation system is established for estimating the AIS signal arrival power at the satellite reception platform. The ionospheric TEC, spatial losses including free space losses, ionospheric scintillation, and Faraday rotation on the power signal are fully considered. The received power is simulated under the existing propagation geometry model as well as antenna conditions. The simulation results are compared with the actual received signal of Tiantuo-5 to verify the simulation system.

## 2. Materials and Methods

### 2.1. Ionospheric TEC

Since the ionospheric TEC varies spatially and temporally, the TEC varies greatly from location to location or from time to time for the same location. In order to more accurately calculate the power loss variation of AIS signals at different times and locations, an accurate global TEC estimation model is needed. In this study, we use the IRI model, which mainly provides important parameters such as electron density, electron temperature, ion composition, ion temperature, and ion drift in the non-polar ionosphere under calm geomagnetic field conditions in the altitude range of 50–1500 km for a given position, time, and date [11,12,13]. Due to the fact that the Tiantuo-5 satellite is moving at an altitude of about 500 km, only the profile data below 500 km are accumulated in the calculation of TEC.

### 2.2. Space Depletion

For AIS signals in the process of space propagation, the signal power loss, in addition to the well-known free space propagation loss, also includes the ionospheric scintillation loss, polarization (Faraday rotation) loss [13], etc.

#### 2.2.1. Ionospheric Scintillations

Ionospheric scintillations are created by fluctuations of the refractive index, which are caused by inhomogeneities in the medium. At the receiver, the AIS signal exhibits rapid amplitude and phase fluctuations and modifications to its time coherence properties. Depending on the modulation of the system, various aspects of scintillation affect the system’s performance differently. The most commonly used parameter characterizing the intensity fluctuations is the scintillation index [11] $S_{4}$, defined by Equation (1):(1)$$S_{4}={\left[\frac{<I^{2}>-{<I>}^{2}}{{<I>}^{2}}\right]}^{\frac{1}{2}}$$
where *I* is the intensity of the signal and < > denotes averaging. $S_{4}$ is related to the peak-to-peak fluctuations of the intensity. Scintillation strength can be classified into three levels, as follows in the Table 1:

As one element of link budget calculations, peak-to-peak amplitude fluctuations: $P_{fluc}$ is related to signal loss $L_{S}$ by [11]:(2)$$L_{s}=P_{fluc}/\sqrt{2}=27.5\cdot S_{4}^{1.26}/\sqrt{2}$$

For weak and moderate levels, $S_{4}$ shows a consistent $f^{-v}$ frequency dependence, with *v* being 1.5 for most multifrequency observations. At the strong level, it has been observed that the *v* factor decreases. This is due to the saturation of scintillation under the strong influence of multiple scattering [11]. As it approaches 1, the intensity follows a Rayleigh distribution. Occasionally, $S_{4}$ at the AIS signal frequency may reach values as high as 1.5, by which the AIS signal path loss caused can reach over 33 dB.

#### 2.2.2. Faraday Rotation

Faraday rotation is the rotation of the sense of polarization as an electromagnetic wave passes through a magnetic field in a transparent dielectric [14,15,16,17,18,19].When propagating through the ionosphere, a linearly polarized wave will suffer a gradual rotation of its plane of polarization due to the presence of the geomagnetic field and the anisotropy of the plasma medium [20,21,22,23]. The Faraday rotation Ω can be calculated as follows [21]:(3)$$\Omega =\frac{K}{f^{2}}\overset{h_{s}}{\underset{0}{\int }}Bcos\theta sec\psi Ndh=K\bar{M}N_{T}/f^{2}$$
where $N_{T}$ is the total electron content on the signal path in electrons/m^2^, *f* is the wave frequency in Hz, *N* is the electron density in electrons per m^3^, *B* is Earth’s magnetic field in Tesla, θ is the angle between the magnetic field and the direction of propagation in radians, ψ is the zenith angle of the satellite in radians, and $h_{s}$ is the height of the satellite in m. For the AIS, the signal is a vertically polarized wave and *f* = 162 MHz. The typical Faraday rotation values in the AIS signal vary from approximately 7.6 rad to 762.1 rad, and for the TEC, from 10 to 1000 TECU (1 TECU = 10^16^ el/m^2^).

As a result of Faraday rotation, polarization loss is caused by the polarization effect on the AIS signal [24,25,26,27,28,29,30,31,32]:(4)$$L_{r}=-10log\left(cos^{2}\alpha \right)$$

The typical polarization loss values of the AIS signal are shown in Figure 1.

### 2.3. Simulation Results

Relative geometry between the satellite and each watercraft was calculated to yield all the required distances and angles in addition to the position of the ionospheric piercing point(IPP). The IPP position was then used to obtain TEC and magnetic field parameters. Path losses and contributions from antenna radiation patterns were calculated and received power at the satellite, and $P_{r}$ was determined:(5)$$P_{r}=P_{t}G_{t}G_{r}L_{mt}L_{mr}\frac{\lambda^{2}}{{\left(4\pi R\right)}^{2}}\frac{1}{L_{p}}$$
where $P_{t}$ is the transmitter power, $G_{t}$ and $G_{r}$ are transmitter and receiver gains, $L_{mt}\;\;\mathrm{and}\;\;L_{mt}$ are the feeder losses at the transmitter and receiver, respectively, and $L_{P}$ is the other losses on the propagation link. Aimed at the AIS signal, the main path loss is from ionospheric scintillation loss $L_{s}$ and polarization loss $L_{r}$:(6)$$L_{p}=L_{s}L_{r}$$
where *R* is the path distance, and *f* is the frequency [4]. The received power of the AIS signal is calculated, respectively, when the ionosphere is quiet and strong ionospheric scintillation occurs, by which the satellite altitude is 500 km:

As shown in Figure 2, when the ionosphere is in a quiet state, the received power at different elevation angles varies as shown in the blue curve, but when a strong ionospheric scintillation occurs, the received power at different elevation angles varies as shown in the red curve.

## 3. Results

The AIS signal observation data is obtained from the satellite Tiantuo-5 designed by the National University of Defense Technology, and the information of receiving time, ship number, position, power, and antenna number is extracted from the data [33,34,35], and the satellite’s corresponding moment position is calculated based on the orbit data. Figure 3 shows the processing flow of the observation data. According to the geometric position between the two, calculate the antenna gain, and finally, calculate the actual power of different positions to reach the satellite.

### 3.1. Satellite Position Calculation

The TLE file, created by the American Celestrak, is an expression used to describe the position and velocity of a spacefaring body, and the orbit of the Tiantuo-5 satellite is calculated using the TLE file. The orbital data of the Tiantuo-5 satellite is given in Table 2.

‘TT-5’ is the satellite name abbreviation; the first row ‘46234U’ indicates that the satellite is numbered 46234, unclassified target (‘U’); ‘20058C’ indicates the 58th launch in 2020; and ‘20246.85751152’ indicates that this orbital root number corresponds to the 246.85751152th day in 2020. The main orbital root numbers are recorded in the second row, and the table below shows the given main orbital root numbers as shown in Table 3.

The calculation is based on the above parameters:

The semimajor axis of the satellite orbit: 6872 km.

Satellite operation cycle: 94.4849 min. Figure 4 shows Tiantuo-5 satellite 3D orbit map and Figure 5 shows Tiantuo-5 satellite star lower point trajectory (one day).

### 3.2. Ship Position Calculation

The distribution of ship and satellite positions during the actual data collection is plotted based on Tiantuo-5 satellite reception data and satellite orbit data. The image in Figure 6 shows the distribution of ships based on AIS data from 12–15 September 2020.

Among them, ‘2020.09.15 03:13:21–04:52:58’ represents the data collection time period, and ‘unit 1’ indicates the receiver number that receives the data. The blue origin indicates the location of the ship, the yellow curve represents the trajectory of the lower point of the satellite star, and the red arrow indicates the direction of the lower point of the satellite star. As seen from the figure, the ships are mainly distributed along the main waterways and coastal areas around the world, and the distribution density varies greatly in different regions.

### 3.3. Geometric Model of Signal Propagation

The relative motion between the ship and the satellite causes the relative position change, and the antenna gain change brought by the relative position change makes us establish two coordinate systems centered on the ship and the satellite in Figure 7.

Where the satellite coordinate system takes the satellite center as the origin, the line connecting the satellite to the center of the Earth as the positive direction of the *Z*-axis, and the satellite flight direction as the positive direction of the *X*-axis. Using the right-hand rule, let the four fingers of the right hand turn 90 degrees from the positive direction of the *Z*-axis to the positive direction of the *X*-axis. At this point, the direction that the thumb is pointing is the positive direction of the *Y*-axis [31,32,33]. The following is the satellite coordinate system.

### 3.4. Antenna Gain Calculation

The satellite-based AIS signal receiver uses dual antennas to receive signals. Two AIS antennas are mounted in the +x and -x planes at 45° to the *z*-axis. AIS antenna 1 is in the +x plane and AIS antenna 2 is in the -x plane. Please see Figure 8.

The AIS antenna is a passive antenna, which is an important part of the AIS receiving system [31]. Table 4 shows main technical indicators. It is mainly composed of a spring radiator, feed connector base, radome, etc., see Figure 9. The spring radiator receives the space electromagnetic wave, converts it into an RF signal, and outputs it through the connector. 

The beam gain also changes with the antenna installation direction because the antenna installation direction is obtained by rotating the antenna 45° around the *Y*-axis in the vertical installation direction. The gain matrix follows the antenna rotation process. Firstly, the 3D polar coordinate system needs to be converted to the right angle coordinate system for matrix rotation. Secondly, the right-angle coordinate system is rotated 45° forward and backward around the Y-axis. Finally, the right-angle coordinates are converted to polar coordinates for the subsequent determination of the beam gain according to the elevation and azimuth angles.
(7)$$\left\{\begin{array}{c} x=rsin\theta \rho \cos\varphi \\ y=rcos\theta \cos\varphi \\ z=r\sin\varphi \end{array}\right.$$

The polar coordinate system is converted to a right-angle coordinate system. Where, r is the gain of the antenna in that direction, φ is the pitch angle of the ship relative to the satellite, and θ is the azimuth angle of the ship relative to the satellite.
(8)$$\left[\begin{array}{c} x^{\prime }\\ y^{\prime }\\ z^{\prime } \end{array}\right]=\left[\begin{array}{ccc} cos\beta & 0 & sin\beta \\ 0 & 1 & 0\\ -sin\beta & 0 & cos\beta \end{array}\right]\left[\begin{array}{c} x\\ y\\ z \end{array}\right]$$

The matrix is rotated around the *Y*-axis, where $\left[\begin{array}{c} x\\ y\\ z \end{array}\right]$ is the matrix before rotation and $\left[\begin{array}{c} x^{\prime }\\ y^{\prime }\\ z^{\prime } \end{array}\right]$ is the matrix after rotation.
(9)$$\left\{\begin{array}{c} r^{\prime }=\sqrt{{x^{\prime }}^{2}+{y^{\prime }}^{2}+{z^{\prime }}^{2}}\\ \theta^{\prime }=arctan\frac{y^{\prime }}{x^{\prime }}\\ \varphi^{\prime }=\arcsin{\left(\frac{z^{\prime }}{r^{\prime }}\right)}^{\;} \end{array}\right.$$

The matrix is converted from the right-angle coordinate system to the polar coordinate system. where $r^{\prime }$ is the gain of the antenna in that direction after rotation, $\varphi^{\prime }$ is the elevation of the ship relative to the satellite, and $\theta^{\prime }$ is the azimuth angle of the ship relative to the satellite [33].

The following Figure 10 shows the consideration of the antenna installation direction to the antenna gain after the influence of the three-dimensional antenna gain graph. Where Figure 10a is for the forward rotation of a 45° antenna beam gain graph, and Figure 10b is for the backward rotation of a 45° antenna beam gain graph. As seen in the figure, after the antenna tilt, the gain of different azimuth angles is no longer the same.

### 3.5. Effect of Satellite Motion

The satellite makes a reciprocal orbit, and the satellite motion coordinate system has a certain angular deviation from the local station center coordinate system, and the deviation is always changing, see Figure 11.

As shown in Figure 11, the XYZ coordinate system is the satellite motion coordinate system (the establishment of this coordinate system is described in detail in Section 2.2). The X′Y′Z′ coordinate system is the local station center coordinate system, with the station center (satellite) as the coordinate system origin O. The Z′ axis is coincident with the Earth’s normal, upward is positive (sky direction), Y′ points to the north direction, and the X′ axis points to the east direction. The angle between the direction of satellite motion and the X′Y′ plane is the elevation deviation, and the angle between the satellite motion and the Y′ axis is the azimuth deviation. The elevation and azimuthal deviations change cyclically with the motion of the satellite, as shown in Figure 12.

To adjust the angular deviation caused by the satellite motion through the rotation of the coordinate system. The azimuthal deviation is first eliminated by rotating counterclockwise around the Z-axis with the origin as the center of the circle (azimuthal deviation).
(10)$$\left[\begin{array}{cccc} x & y & z & 1 \end{array}\right]=\left[\begin{array}{cccc} X & Y & Z & 1 \end{array}\right]\left[\begin{array}{cccc} cos\theta & sin\theta & 0 & 0\\ -sin\theta & cos\theta & 0 & 0\\ 0 & 0 & 1 & 0\\ 0 & 0 & 0 & 1 \end{array}\right]$$

Then, elevation deviation elimination is carried out by rotating counterclockwise around the *X*-axis by degrees (elevation deviation) with the origin as the center of the circle [32,33].
(11)$$\left[\begin{array}{cccc} X^{\prime } & Y^{\prime } & Z^{\prime } & 1 \end{array}\right]=\left[\begin{array}{cccc} x & y & z & 1 \end{array}\right]\left[\begin{array}{cccc} 1 & 0 & 0 & 0\\ 0 & cos\varphi & sin\varphi & 0\\ 0 & -sin\varphi & cos\varphi & 0\\ 0 & 0 & 0 & 1 \end{array}\right]$$

## 4. Discussion

The measured data of the ship in Section 3.1 are processed, and the results are compared with the simulated power. The variation of the measured power (blue) and simulated data (red) with elevation is given in the Figure 13.

From the above results, it can be calculated that the root mean square between the simulated and measured power of AIS on Tiantuo-5 is around 1 dB, indicating that the simulation data and the measured data fit well.

## 5. Conclusions

In this research, a simulation model was established to assess the performance of an AIS receiver on the Tiantuo-5 satellite. The model takes into account neutral atmosphere and ionospheric effects, ship-satellite geometry, and antenna radiation patterns. The received signal power of AIS was calculated when the ionosphere is quiet and strong ionospheric scintillation occurs, respectively. We found that the maximum AIS signal path loss caused by ionospheric scintillations can reach over 33 dB. The Faraday rotation caused by the ionosphere has a severe impact on the received power of satellite-based AIS receivers. The polarization loss caused by the Faraday rotation of the AIS signal changes rapidly with the change of the Faraday rotation, and the polarization loss can reach over 35 dB when the Faraday rotation is close to a multiple of 90 degrees.

To verify the effectiveness of the established simulation model, the real measured power of the receiver on the Tiantuo-5 satellite is processed, and the results are compared with the simulated power. As the antenna gain changes as the relative position of the satellite and ship changes, the orbit of the Tiantuo-5 satellite was calculated by the TLE file, the position of those ships was decoded from the AIS message, and two coordinate systems centered on the ship and the satellite were established in order to obtain the antenna gain in different ship-satellite geometry situations. The root mean square error between the simulated and measured power of AIS on Tiantuo-5 is around 1 dB, indicating that the simulation data and the measured data fit well. This simulation model can be used for the performance assessment of the satellite-based AIS receivers and the design of the available power and antenna of the satellite-based AIS receivers.

However, due to the uneven distribution of AIS signals around the world and the fact that the power of AIS signals and antenna gain vary from ship to ship [36,37], the technique used in this paper can only yield a statistical result, and a more accurate power analysis is not possible. To overcome this disadvantage, the team is working on building our own AIS signal transmitting equipment. This will allow us to analyze the entire link more accurately based on the precise technical parameters of the transmitted signals and thus find even more minute deficiencies in the algorithm.

## Figures and Tables

**Figure 1 sensors-23-06740-f001:**
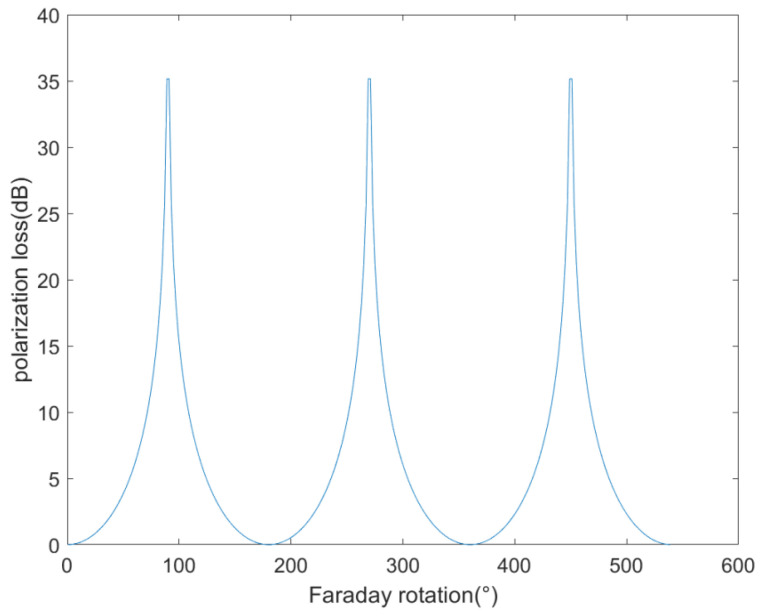
AIS signal polarization loss with ionospheric TEC.

**Figure 2 sensors-23-06740-f002:**
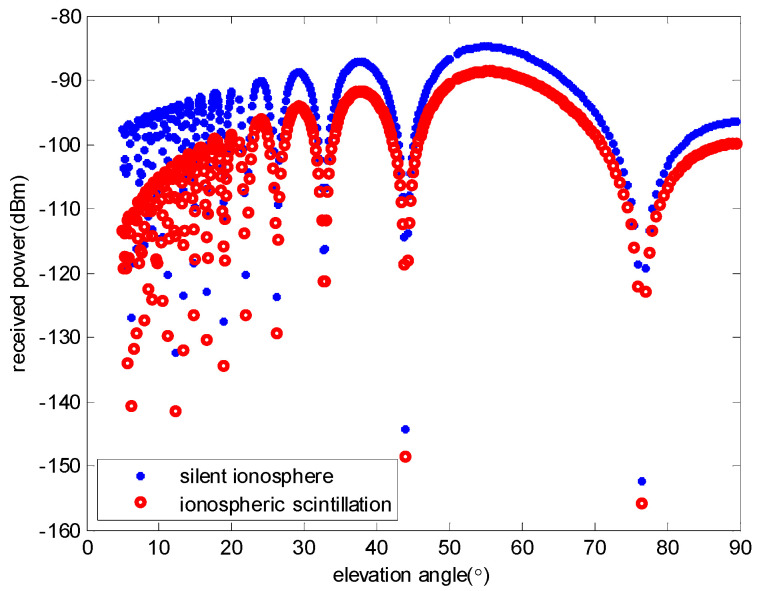
Received power of the AIS signal.

**Figure 3 sensors-23-06740-f003:**
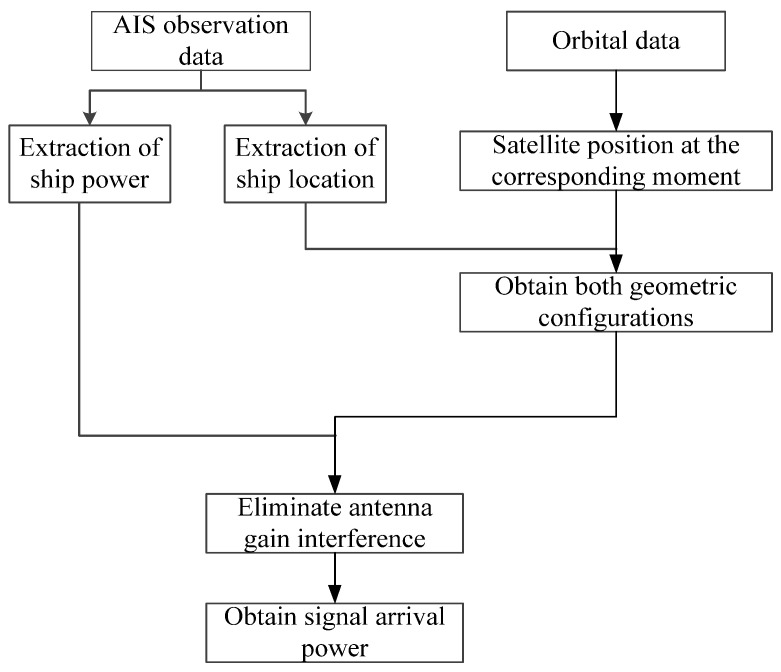
Observation data processing float.

**Figure 4 sensors-23-06740-f004:**
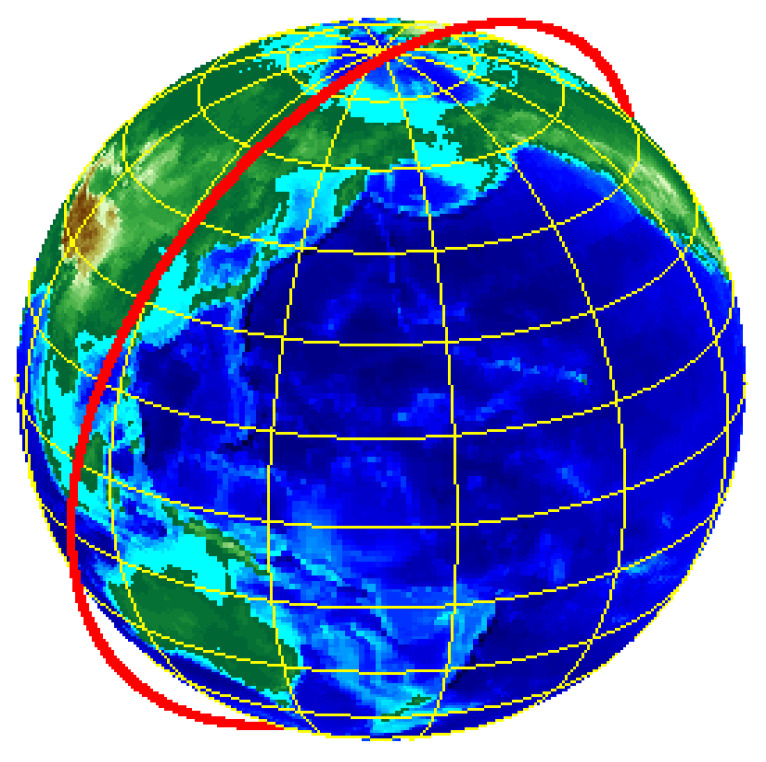
Tiantuo-5 satellite 3D orbit map.

**Figure 5 sensors-23-06740-f005:**
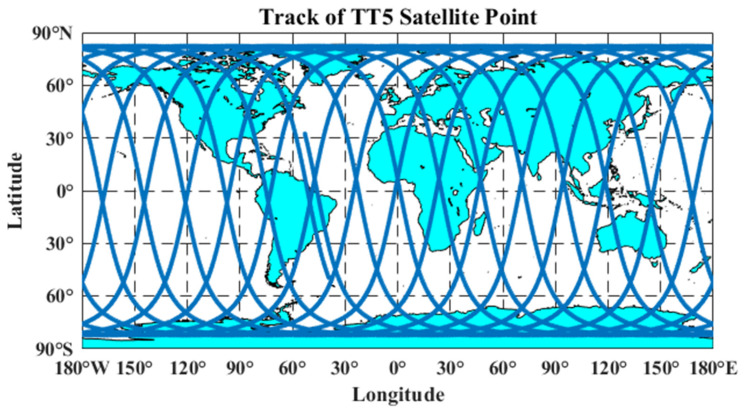
Tiantuo-5 satellite star lower point trajectory (one day).

**Figure 6 sensors-23-06740-f006:**
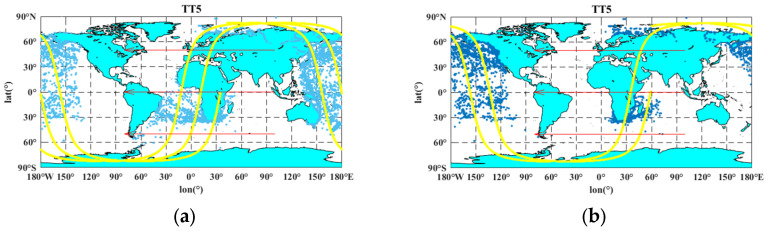
Distributions of ships. (**a**) 2020.09.12 06:28:06–10:48:22, unit 1; (**b**) 2020.09.12 04:52:59–07:40:46, unit 2; (**c**) 2020.09.12 08:00:36–15:37:27, unit 2; (**d**) 2020.09.13 02:23:00–03:49:22, unit 1; (**e**) 2020.09.13 02:21:23–03:52:15, unit 2; (**f**) 2020.09.14 19:29:32–19:39:34, unit 1; (**g**) 2020.09.13 13:25:43–15:17:03, unit 2; (**h**) 2020.09.14 07:14:12–13:28:29, unit 2; (**i**) 2020.09.14 16:39:30–19:21:40, unit 1; (**j**) 2020.09.15 03:13:21–04:52:58, unit 1.

**Figure 7 sensors-23-06740-f007:**
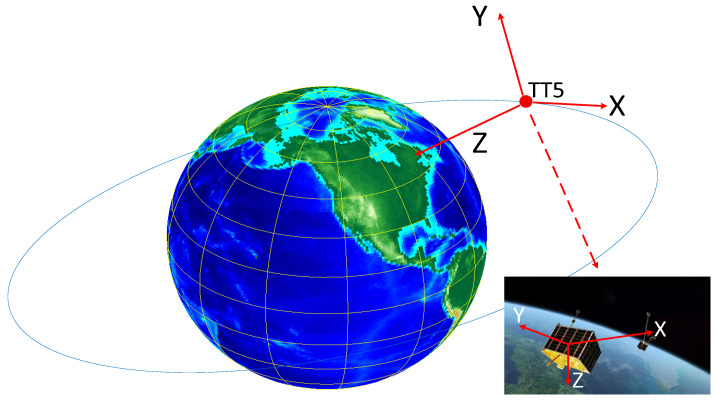
Distributions of the schematic diagram of the coordinate system.

**Figure 8 sensors-23-06740-f008:**
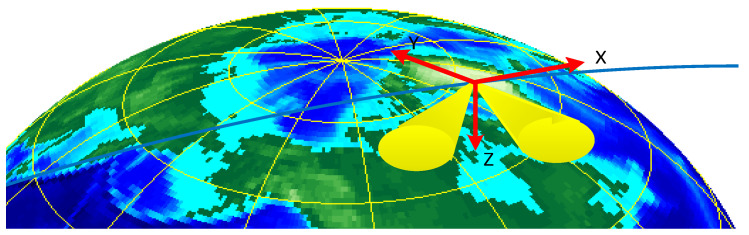
AIS antenna distribution.

**Figure 9 sensors-23-06740-f009:**
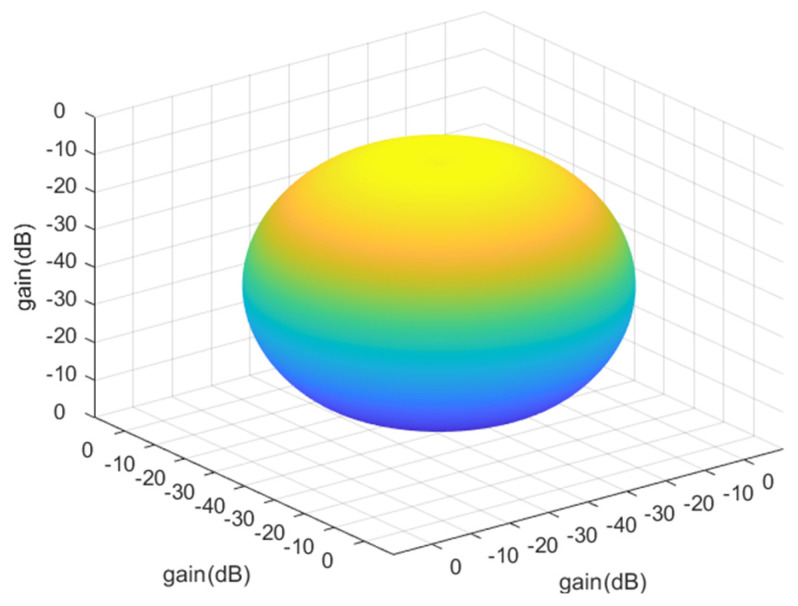
Three-dimensional beam pattern.

**Figure 10 sensors-23-06740-f010:**
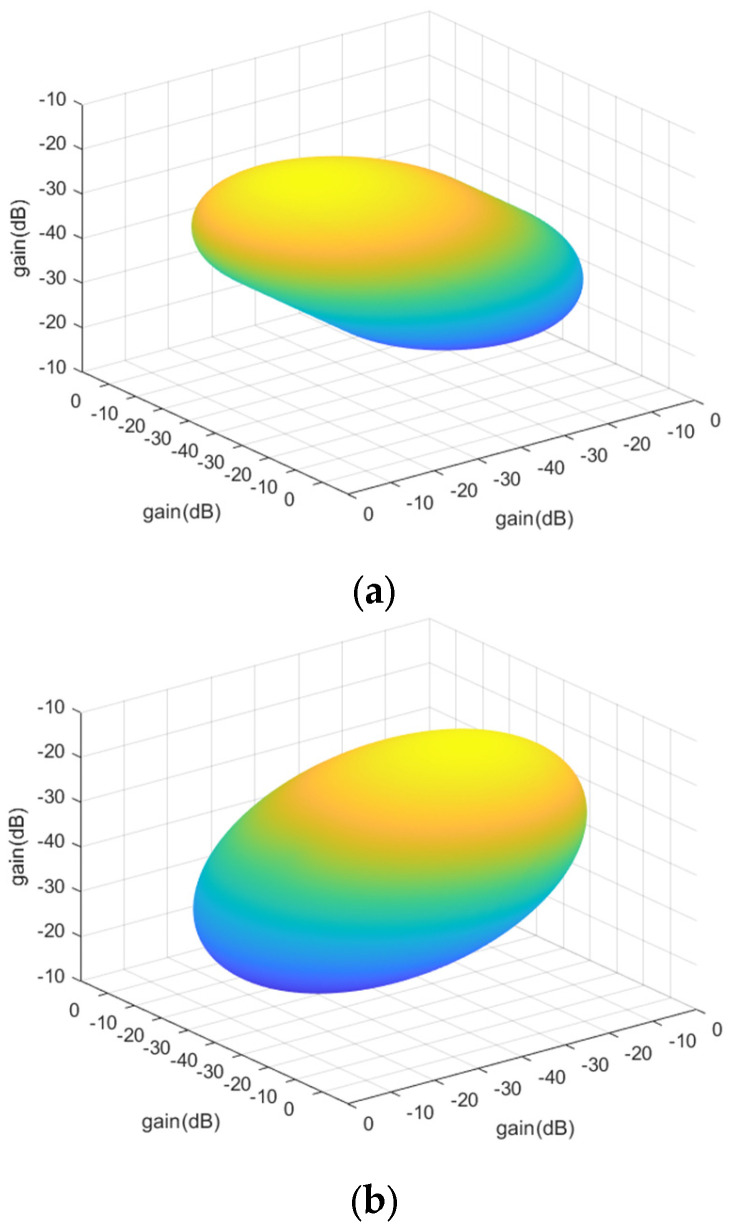
Three-dimensional beam pattern. (**a**) Forward rotation 45°; (**b**) Backward rotation 45°.

**Figure 11 sensors-23-06740-f011:**
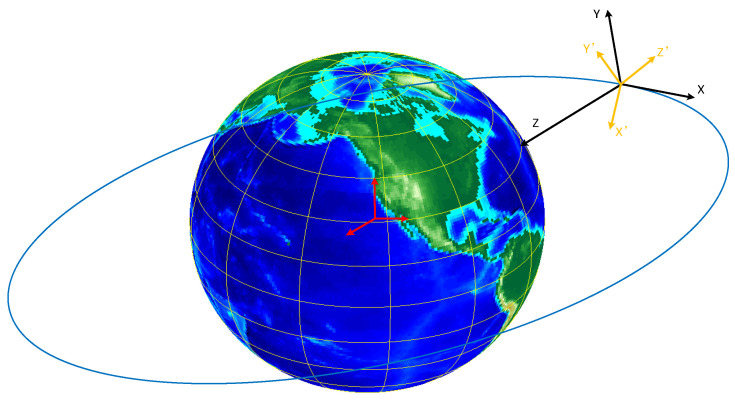
Coordinate system.

**Figure 12 sensors-23-06740-f012:**
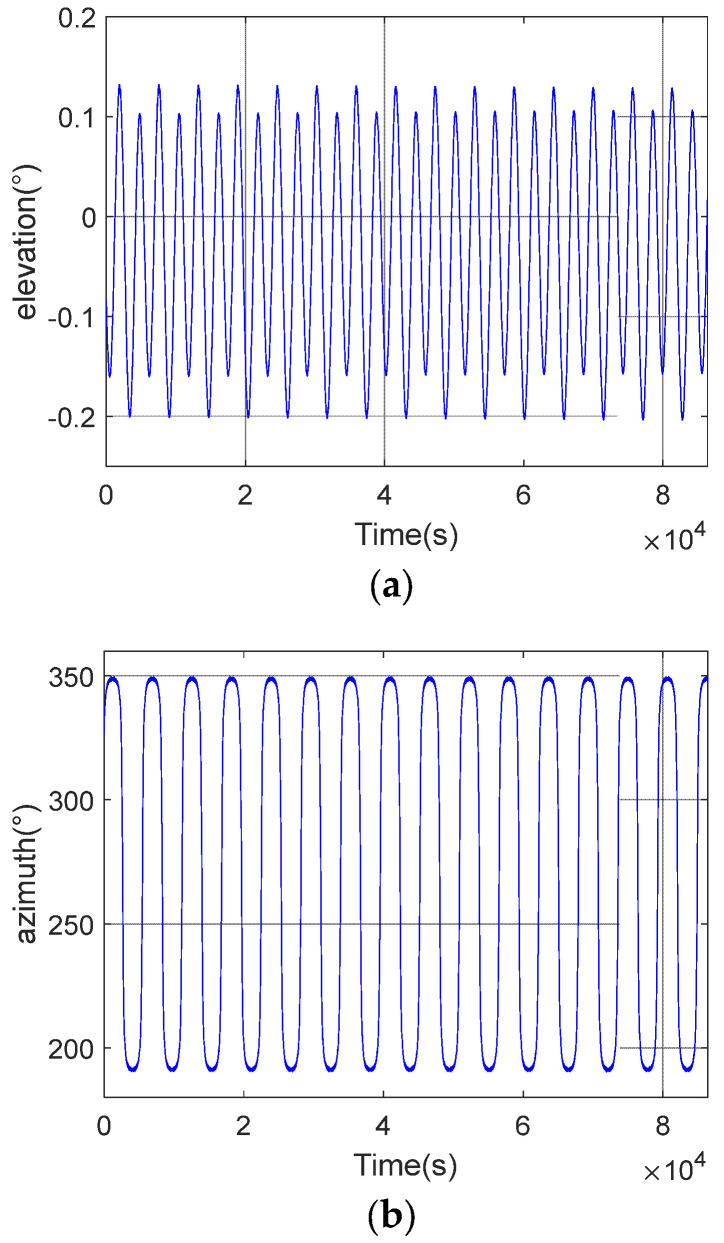
Change of the motion direction. (**a**) Elevation change; (**b**) Azimuth change.

**Figure 13 sensors-23-06740-f013:**
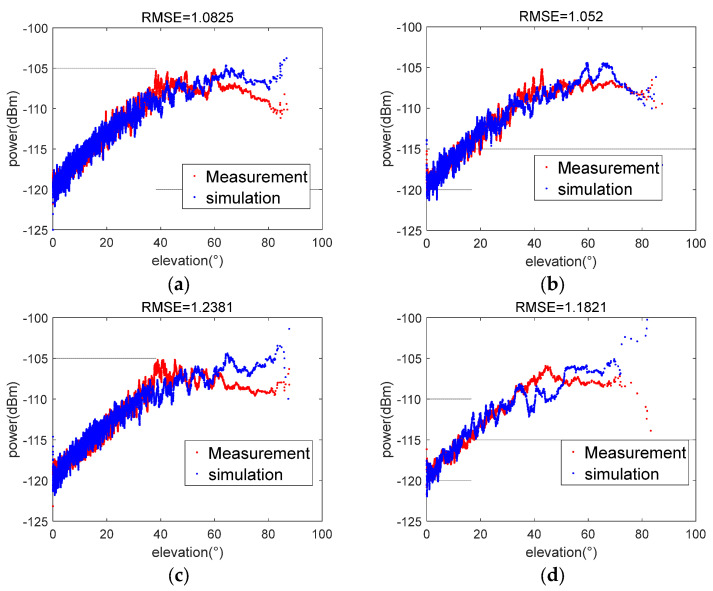
Comparison of the simulated and measured data. (**a**) 2020.09.12 06:28:06–10:48:22; (**b**) 2020.09.12 04:52:59–07:40:46; (**c**) 2020.09.12 08:00:36–15:37:27; (**d**) 2020.09.13 02:23:00–03:49:22; (**e**) 2020.09.13 02:21:23–03:52:15; (**f**) 2020.09.14 19:29:32–19:39:34; (**g**) 2020.09.13 13:25:43–15:17:03; (**h**) 2020.09.14 07:14:12–13:28:29; (**i**) 2020.09.14 16:39:30–19:21:40; (**j**) 2020.09.15 03:13:21–04:52:58.

**Table 1 sensors-23-06740-t001:** Ionospheric scintillation strength levels.

Scintillation Level	Weak	Moderate	Strong
$S_{4}$	<0.3	0.3–0.6	>0.6

**Table 2 sensors-23-06740-t002:** TLE file.

TT5								
1	46234U	20058C	20246.85751152	0.00000202	00000-0	11337-4	0	9995
2	46234	97.5043	294.3731	0015061	229.4703	185.1211	15.24052466	1621

**Table 3 sensors-23-06740-t003:** Orbit parameters.

Num.	Parameter	Value
1	Satellite number	46234
2	Inclination	97.5043
3	RAAN	294.3731
4	Eccentricity	0.0015061
5	Argument of perigee	229.4703
6	Mean anomaly	185.1211
7	Number of circles around the Earth per day	15.24052466
8	Number of laps flown since launch	1621

**Table 4 sensors-23-06740-t004:** Main technical indicators.

Num.	Indicators	Company	Value
1	frequency	MHz	156.8 ± 0.5; 162 ± 0.5
2	gain	dBi	≥−1 (maximum gain)
3	standing wave	/	≤2@156.8 MHz; ≤1.5@162 MHz
4	impedance	Ω	50
5	polarization form	/	linear polarization

## Data Availability

The data set is available on request to the corresponding authors.

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
