# Peer review of "Comparison of Simulated and Measured Power of the Earth-Space Link for Satellite-Based AIS Signals"

_sensors, 2023, doi:10.3390/s23156740_

Round 1

Reviewer 1 Report

The topic is interdisciplinary, with a great potential for practical implementation. However, several elements could be updated in order to raise the quality of this manuscript.

Suggestions and comments:

1) To start with, this paper is quite short, most of its contents include mathematical equations and figures. Authors are strongly advised to extend the main text, particularly: introduction, related works, feedback from own findings, conclusions and comments. However, if this is intended to be a [communication] paper, and it meets all necessary requirements, it is good to go.

2) In the paper title, the words Measured and Space should be written using Capital Letters.

3) In the whole text, the word Earth should be written with a Capital E, as it refers to our Planet. A similar remark goes to Faraday, since it comes from the name of the Inventor.

4) The number and type of Keywords should be reconsidered.

5) Notice and modify how to properly insert citations in text.

6) All acronyms/abbreviations appearing in text should be fully described when first mentioned.

7) Several minor editorial and formatting issues are present, therefore a careful throughout examination would seem necessary.

8) Mathematical symbols as well as equations appearing in text do not seem to be properly edited.

9) The number and scope of cited references is very limited. Authors are strongly advised to extend their range. Furthermore, they do not seem to be properly and uniformly formatted. Carefully check the journal template and editorial requirements and make necessary corrections.

10) Figure 4 should be redone, as the bars are not necessary.

11) Table 2 – the Remarks column is left empty. Shouldn’t there be any kind of information.

12) What kind of laboratory stand was used during the study, including the hardware (PC components) and software (simulation environment, libraries, toolboxes, etc.). At least principle information should appear. Additionally, what kind of antenna array and related equipment, etc., was utilized to obtain this data? What kind of frequency band did you examine?

13) Figure 6 should be inserted in higher resolution and/or different file format.

14) Figure 7 is simply too small, the fonts are not legible. Consider presenting it in larger size or a different layout.

15) Similar remarks go to Figures 13 and 14.

16) How long did the experiment last – one day, a week, or maybe the whole month?

17) The Conclusions section is far too short and not convincing at all. Do provide additional feedback on your findings, mention open issues and future study directions.

18) Chapter 6 is blank, delete it if it is not necessary.

To sum up, this paper requires a major revision before it can be processed further.

3) In the whole text, the word Earth should be written with a Capital E, as it refers to our Planet. A similar remark goes to Faraday, since it comes from the name of the Inventor.

7) Several minor editorial and formatting issues are present, therefore a careful throughout examination would seem necessary.

Reviewer 2 Report

Dear authors,

I have a question:

About the Faraday rotation, did you thinking about the using of the antennas with circular polarisation?

and please solve the following:

The variables in the text are flying up the other letters. (sometimes also the references are up the other text)

Add the source of the equation 3

Equation 5 you should add, that the units are in dB, and I suggest to add the better one equation, used one looks like for the bachelor students (it doesn't matter if in dB or Watts): P_r = P_t * G_t * G_r * f_r^2(fi_1,psi_1) * f_t^2(fi_2, psi_2)*(lambda/(4*pi*R))^2 * 1/(Losses)

Lines 134-137: add how many percent is in the selected intervals, nice can be distribution function.

figure 3: paint the algorithm better (look on the wires connection for example here: https://en.wikipedia.org/wiki/Algorithm#/media/File:Euclid_flowchart.svg)

Why the table 2 has column remarks, it is empty, so remove it.

figure 10: move the antenna on the position [0,0,0], (see: https://www.industrialnetworking.com/pdf/Antenna-Patterns.pdf)

Figure 11 seems to be weird, if you didn't modified shape, the beam pattern must be same just depends  the gain on the angle (see equation 5 I wrote you). If you rotate the ball shape, it must stay to be ball shape. I think that there should be just selected the point on the original pattern.

line 294: where is zero (0)?

Describe better the situation around the figure 14, now it is looking that if you are monitoring in the main lobe you are receiving the weakest signal.

Best Regards.

Round 2

Reviewer 1 Report

Authors have prepared a revised version of their initial submission. Now the paper is more informative and understandable for the potential reader. The topic is practical and up to date, it is surely worth presenting and publishing in the Journal. There are still some editorial and formatting issues, in the main body of the manuscript as well as list of cited references. However, all of them can be overcome at a later stage.

To sum up, in my opinion this paper meets necessary requirements in order to be accepted and published.

Author Response

Point 1:Authors have prepared a revised version of their initial submission. Now the paper is more informative and understandable for the potential reader. The topic is practical and up to date, it is surely worth presenting and publishing in the Journal. There are still some editorial and formatting issues, in the main body of the manuscript as well as list of cited references. However, all of them can be overcome at a later stage.

To sum up, in my opinion this paper meets necessary requirements in order to be accepted and published.

Response 1:  Many thanks for this comment. We have checked the hole paper.

Reviewer 2 Report

Dear authors,

Check the lg in equation (4)

Equation 5, use the description I sent you last time, if you want to talk about specific part, just highlight this part of the equation.

About the figure 6, if you connect two ways/directions (arrows), there should be full dot and arrows, we are not in the nineties, make it visually better.

Discuss, why the received power from the main direction is weaker (almost 15 dB) than the power received from the 60°? (the reason for the major revision, explain it carefully)

Check that the titles are on the same page as the figure or table. It is better if the table is not spread on the more pages, if it is necessary, put there again the head line.

Best Regards.

note: please, use an unique color in the manuscript for the answers to my questions (for simplier checking).

Author Response

Point 1:Check the lg in equation (4)

Response 1: Many thanks for this comment.We've made a correction in the article.

Point 2:Equation 5, use the description I sent you last time, if you want to talk about specific part, just highlight this part of the equation.

Response 2: We have made changes to the images in the appropriate places, see the corresponding section of the article for details of the changes.

Point 3:About the figure 6, if you connect two ways/directions (arrows), there should be full dot and arrows, we are not in the nineties, make it visually better.

Response 3:We have made changes to the images in the appropriate places, see the corresponding section of the article for details of the changes.

Point 4:Discuss, why the received power from the main direction is weaker (almost 15 dB) than the power received from the 60°? (the reason for the major revision, explain it carefully)

Response 4:In order to more accurately compare the simulation results with the measured results, we removed the power differences caused by the antenna beam gain from the measured data during data processing. The power loss results thus formed are only correlated with the spatial propagation links.So the comparison between simulated and measured results in Fig. 13 is independent of the antenna beam gain.

Point 5:Check that the titles are on the same page as the figure or table. It is better if the table is not spread on the more pages, if it is necessary, put there again the head line.

Response 5:Many thanks for this comment.We've made a correction in the hole article.